# Efficacy and Short-Term Safety of COVID-19 Vaccines: A Cross-Sectional Study on Vaccinated People in the UAE

**DOI:** 10.3390/vaccines10122157

**Published:** 2022-12-15

**Authors:** Mustafa Ameen AlHamaidah, Noora Noureldin, Adham Yehia, Ibrahim Alani, Abdelaziz Al-Qussain, Osama Abdou, Akram Ashames, Zelal Kharaba

**Affiliations:** 1Department of Pharmaceutical Sciences, College of Pharmacy and Health Science, Ajman University, Ajman P.O. Box 346, United Arab Emirates; 2Medical and Bio-Allied Health Sciences Research Centre, Ajman University, Ajman P.O. Box 340, United Arab Emirates; 3Department of Clinical Pharmacy, College of Pharmacy, Al Ain University, Abu Dhabi 112612, United Arab Emirates; 4AAU Health and Biomedical Research Center, Al Ain University, Abu Dhabi 112612, United Arab Emirates; 5Faculty of Medical Sciences, Newcastle University, Newcastle Upon Type NE2 4HH, UK

**Keywords:** COVID-19 vaccines, safety, efficacy, side effects, United Arab Emirates

## Abstract

Background: The emergence of COVID-19 has been a major challenge to public health and the world economy. During a wave of COVID-19, the usage of widespread vaccination procedures and broader coverage to the whole of humanity will be made possible if the general population has access. An intended effect of vaccination is to provide “herd immunity,” which protects those who have not been vaccinated along with those who have been. However, some concerns regarding the safety and efficacy of COVID-19 vaccines were raised. Aim: This study aims to provide evidence on the short-term safety and efficacy of four types of vaccines that are officially approved by the Ministry of Health in the United Arab Emirates (UAE). These include Sinopharm, Sputnik V, Pfizer, and AstraZeneca. Method: This study utilized a cross-sectional descriptive design. Data on the efficacy and short-term protection of COVID-19 vaccines on vaccinated citizens and residents (*n* = 764) of the UAE were collected between February and April 2021. Participants were conveniently approached using a Google Forms survey, where they responded to a semi-structured questionnaire pertaining to socio-demographic questions and in-depth questions related to COVID-19, including whether they suffer from any comorbidities, the most commonly encountered post-vaccination side effects, and the severity of their symptoms, using a 5-point Likert scale. Results were analyzed using SPSS version 24, calculations of *p*-values and descriptive statistics were used for data differentiation. Results: The majority of the participants (*n* = 612 or 94.4%) stated that they did not become reinfected after receiving two doses of COVID-19 vaccine. In addition, the incidence of being hospitalized after vaccination was negligible. In terms of adverse effects, the most common individually reported side effects, regardless of the vaccination type, included “pain at the site of injection”, followed by “general fatigue”, then “lethargy”. Moreover, most of these side effects occurred after the second dose of the vaccine, irrespective of the type of vaccine. Females were found to be more susceptible to the adversities of COVID-19 vaccination. The occurrence of side effects was not found to be related to the nationality/ethnicity of the vaccine recipient. Furthermore, none of the vaccines affected sleep pattern, since a significant number of respondents reported a regular sleep pattern after being vaccinated. The majority respondents who received two doses of vaccination (*n* = 585 or 76.6%) reported that they did not become infected post vaccination, regardless the type of vaccine received, whereas only (*n* = 11 or 1.9%) were reinfected with COVID-19 after 2–4 weeks. Conclusion: The findings of this study suggest that vaccines can offer short-term protection against COVID-19 reinfection. Moreover, both the first- and second-vaccination side effects were described as very mild to moderate, which indicates tolerability. These data may strengthen the public confidence in receiving vaccinations.

## 1. Introduction

The commencement of the global lockdown in 2020, which was enforced due to the COVID-19 pandemic, has given a rise to the emergence effective solutions to combat SARS-CoV-2 [1]. Despite the fact that a lot of things still remain unknown when it comes to the methods by which the SARS-CoV-2 may develop in the long term, the scientific community agrees that the invention of an effective vaccine, and utilizing it globally, could be the best solution to put an end to the COVID-19 pandemic [2]. Ref. [3] proposes that numerous organizations, laboratories, and institutes across the world are researching and developing a vaccine that can provide immunity to protect those who are at risk from infection. According to the New York Times [3], the world total number of vaccines on which researchers are working is 165; approximately 135 coronavirus vaccines are in the preclinical stage (created in laboratories but not yet tested in human trials), 21 have entered Phase I (to be tested for safety and dosage in human trials), 13 have entered Phase II (to be tested in a larger number of human trials), 8 have entered Phase III (to be tested in a larger number of human trials), and yet just 2 are approved for use on immunocompromised individuals. According to the author Turak [4], a family travelling from Wuhan to the UAE on January 16, 2020 were found to be infected with the virus. Consequently, the government took required measures to contain the spread of coronavirus, with a response rate that was faster compared to other countries. All events were canceled, and entertainment venues were shut down. Visa services were suspended for all foreigners starting from 17 March 2020 [5]. According to the National Emergency Crisis and Disasters Management Authority, the total number of individuals diagnosed with COVID-19 in the UAE from the beginning of the pandemic until 21 October 2022 is 1,034,462, with 2348 deaths registered due to this infection [6].

Clearly, the spread of COVID-19 put a lot of pressure on pharmaceutical companies to develop vaccines to contain the spread of this deadly virus. Thus, scientists from all around the world developed vaccines after multiple trials and a lot of hard work. Although there are various vaccines available globally, this study will only focus on those that have been approved by the Ministry of Health in the UAE: Sinopharm, Sputnik V, Pfizer, and AstraZeneca [7]. The first vaccine is Sinopharm, developed by a Chinese company belonging to the China National Pharmaceutical Group. The Sinopharm/BBIBP-CorV vaccine received approval in December 2020 for general public trials, and according to those trials, it showed a success rate of 79% against SARS-CoV-2 infection [8]. Phase III clinical trials of the BBIBP-CorV vaccine consisted of 45,000 volunteers, and took place in June 2020 in seven countries: Argentina, Peru, Jordan, Bahrain, Egypt, Morocco, and the UAE [9]. A total of 31,000 volunteers participated in the Phase III clinical trials across all the UAE, with only 15,000 individuals from the emirate of Abu Dhabi [10]. Preliminary results of this trial encompassing the UAE and Bahrain, with a total number of 40,411 volunteers who received inactivated forms of the vaccines from two dissimilar viral strains (i.e., WIV04 and BBIBP-CorV), revealed an efficacy rate of 72.8% (for WIV04) and 78.1% (for BBIBP-CorV) in preventing symptomatic cases [10]. Additionally, it was reported that these vaccines offered 99% rate of seroconversion, and 100% protection against severe COVID-19 infection [10]. A very recent retrospective cohort study conducted in the emirate of Abu Dhabi in the UAE revealed that the efficacy of the Sinopharm vaccine against hospitalization is 79.6% (95% CI, 77.7 to 81.3), whereas the efficacy against critical care unit admission was found to be 86% (95% CI, 82.2 to 89.0), with only 84.1% (95% CI, 70.8 to 91.3) effectiveness rate against death due to SARS-CoV-2 infection [11]. Moreover, the study results pointed out that the efficacy of this vaccine against severe COVID-19 outcomes has declined over the time, which highlights the importance of administering booster doses to improve the vaccine’s protective capacity [11].

The second vaccine is Pfizer, an RNA-based vaccine that was developed by the German company BioNTech, which was originally an American pharmaceutical company [12].

The BNT162b2 and BNT162b1 vaccines are mRNA-based vaccines. They are formulated via the injection of a synthetic mRNA into a protein, and then translated rapidly by the host cell [13]. Treatment using mRNA technology is considered well-tolerated and safe due to the rapid metabolism and the transient expression of the RNA, as well as the avoidance of integration into the host genome [14]. Recently, it has been found that modifying mRNA molecules with 1-methylpseudouridine has given a rise to a long-term antibody response, whereas encompassing the mRNA with liquid nanoparticles offered protection against degradation [15]. Back in December 2020, BNT162b2 received authorization for emergency use, temporarily, in the UK, based on data submitted from Phase III clinical trials; consequently, a series of authorization approvals for emergency use took place in other countries, including Mexico, Canada, USA, Bahrain, and Saudi Arabia [16]. BNT162b2 requires storage in a temperature range of −80 °C to −60 °C, and it must be thawed, then diluted, prior to use [17], which represents a challenge for vaccine distribution [18]. The efficacy rate of BNT162b2 was reported as 95.0% effective (95% CI 90.3–97.6) in preventing SARS-CoV-2 infection in patients without a prior history of contracting COVID-19, up to 7 days after receiving the second dose of the vaccine in Phase II/III clinical trials [19]. Additionally, in global Phase I/II/III placebo-controlled clinical trials (NCT04368728), only 8 cases of SARS-CoV-2 infection were reported, with an onset of more than or equal to 7 days after the second dose in the BNT162b2 recipients group, whereas 162 cases of COVID-19 infection took place in placebo recipients [19]. Mousa et al. [20] conducted a study in the UAE to evaluate the efficacy of mRNA BNT162b2 (Pfizer-BioNTech) and BBIBP-CorV (Sinopharm) against the new variant of COVID-19, the Delta variant (B.1.617.2), among fully vaccinated individuals. Their results demonstrated that Sinopharm vaccine efficacy in preventing critical care hospital admissions was 95% (95% CI: 94, 97%)], while the Pfizer-BioNTech vaccine had an efficacy rate of 98% (95% CI: 86, 99%) [20].

The third vaccine is Sputnik V, a Russian vaccine produced by the Gamaleya National Center of Epidemiology and Microbiology medical research institute in Russia [21]. It is composed of a two-part adenovirus viral vector, which aims to provoke the production of antibodies that counteracts spike proteins [21]. The launch of the Sputnik V vaccine was rather controversial among the scientific community, especially due to the fact that Russia announced their vaccine and approved it in August 2020 prior to gathering detailed clinical data [22]. Moreover, the efficacy rate of 92% claimed upon releasing the initial results was criticized since it was based on a very low number of participants [22]. However, in February 2021, the results of Phase III randomized, placebo-controlled, double blind clinical trials involving 22,000 adults aged 18 years and above were published [23]. Participants in this study either received a placebo or two doses of the vaccine, with a spacing duration of 21 days, and the overall efficacy reported in this paper was 90% among 14,964 vaccinated individuals, and no serious side effects were reported, as most of the adverse events were mild, although over 50% of vaccine recipients experienced pain at injection site [23].

The fourth vaccine is AstraZeneca, which was developed by Oxford University, England, and sold under the names Covishield and Vaxzevria [24]. More than 20 million people were vaccinated in the UK with this vaccine, among which, 79 cases of blood clots and 19 deaths were reported [25]. These numbers equate to around one case of a blood clotting adverse event per 250,000 people vaccinated with the AstraZeneca vaccine, with an incidence rate of 0.0004% and one death in a million [26]. The European Medicines Agency (EMA) reported that “benefits of vaccination outweigh any risks of side effects”, because COVID-19 infection also poses a dangerous risk of developing fatal blood clots, including deep venous thrombosis and pulmonary embolism [27]. Moreover, the EMA stated that there was no ultimate link found between the vaccine and the direct cause of blood clots, and this should rather be described as a rare immune response towards the vaccine [26]. On March 2022, the company announced that the initial analysis of AstraZeneca vaccine efficacy revealed 79% efficacy rate in preventing SARS-CoV-2 infection in a multinational clinical trial that included 32,449 adults from Peru, Chile, and the US [28]. It is also noteworthy that there was no hospitalization or death cases among the participants who received two doses of the vaccine, despite the fact that 60% of them had comorbidities that are linked to increased risk of developing severe symptoms such as obesity or diabetes [28].

Despite the high rate of morbidity and mortality associated with COVID-19, many individuals rejected the vaccination for various reasons. A randomized, cross-sectional study conducted in Jordan, implementing a machine learning approach for predicting the severity of side effects of COVID-19, found that about 45% of individuals feared the side effects, 29% did not trust vaccines at all, and 20% did not even know how vaccines worked [29]. By refusing to vaccinate, individuals are not only harming themselves but the people around them. Since discussing what to expect post vaccination will aid in lowering the community’s trepidation and hesitancy towards the different types of vaccines available, we sought to perform this study to demonstrate the possible short-term adverse effects of COVID-19 vaccines, as well as their short-term efficacy, in order to maximize the trust in the vaccination process and speed up the process of gaining herd immunity.

## 2. Methodology

### 2.1. Study Design and Participants

This is a cross-sectional study, carried out between 23 February 2021 and 1 April 2021, involving citizens and residents of the UAE’s population across all seven emirates. The study was based on a self-administered online survey created via Google Forms. The distribution of the survey among participants was conducted using the random sampling technique with the help of social media platforms (e.g., WhatsApp, Facebook, and Telegram). Individuals willing to participate in the study were redirected to a webpage that included an introduction about the aim of the research, along with an invitation message that allowed them either to decline or accept participating. Then, respondents were asked to sign an e-consent form for their participation, in the knowledge that their participation was strictly voluntary, that all the gathered data would be anonymous, and that they had the right to withdraw from the study at any time.

Overall, the study’s questionnaire reached a total of 770 participants. Of this, 764 participants completed the questionnaire and returned complete data, which corresponds to a 95.14% response rate. This study acquired a confirmation letter from the Research Ethics Committee (REC) of Ajman University on 17 February 2021, and it was approved as a valid cross-sectional study by the REC of the College of Pharmacy under the reference number of PHS-2020-12-3.

### 2.2. Inclusion and Exclusion Criteria

The study population included citizens and residents of the UAE aged >18 years that had received at least one dose of the four previously mentioned COVID-19 vaccines.

### 2.3. Study Tools and Data Collection

The questionnaire of the study was designed and developed by the authors with the help of a thorough review of the literature concerning previous studies on the anticipated adverse events following COVID-19 vaccinations [30,31]. The survey consisted of four sections. The first section included seven demographic items (age group, gender, nationality, educational level, employment status, city of residence, and blood group type). In order to appropriately inspect the efficacy and short-term safety of COVID-19 vaccines in the UAE population, it was necessary to identify the important parameters and variables that might affect the vaccines’ extent of effectiveness and safety. Therefore, in the second section of the survey, we included nine close-ended questions, which aimed to detect the presence of comorbidities or genetic diseases, as well as the concomitant use of specific groups of medications or addictive substances, and the body weight of the participants. Moreover, participants were asked about the type of vaccine, the number of shots received, and the time of receiving the vaccination. The third section of the survey aimed to investigate the efficacy of the COVID-19 vaccines; therefore, the participants were asked whether they had become reinfected with COVID-19 despite receiving two doses of the vaccine. Additionally, they were asked when they caught the infection, and whether they required hospitalization upon reinfection after receiving the vaccine. Moreover, to assess whether the vaccine helped in minimizing the severity of COVID-19 infection symptoms or not, a 5-point Likert scale question about the severity of the infection post vaccination was asked, which is statistically expressed by the standard deviation and by calculating the mean of the respondents’ answers frequencies (out of 5), where (5 = Severe COVID-19 infection symptoms, 4 = Moderate COVID-19 infection symptoms, 3 = Mild COVID-19 infection symptoms, 2 = Very slight COVID-19 infection symptoms, 1 = Not been infected). To explore the safety of the COVID-19 vaccine, we asked the participants, in the fourth section, some investigational questions regarding the side effects they suffered post vaccination, mentioning the following adverse effects: general fatigue, lethargy, muscle pain, pain at site of injection, fever, dizziness, headache or migraine, diarrhea, other side effects, none, itch and rash, malaise, lymphadenopathy, multiple side effects. Moreover, to assess the severity of the experienced side effects of the recipient, a 6-point Likert scale question was asked, which is statistically expressed by the standard deviation and by calculating the mean of the respondents’ answers frequencies (out of 6), where (1 = Very slight side effects, 2 = Mild side effects, 3 = Moderate side effects, 4 = Severe side effects, 5 = Very Severe side effects, 6 = Worst Possibility.

### 2.4. Questionnaire Development and Validation

The principal investigator of the study invited six experts in the clinical pharmacy field, who were professors in clinical pharmacy and pharmacy practice from three different universities in the UAE, as well as five expert infectious disease physicians to attend an online meeting for the aim of validating the content of the survey. The panel members were asked to grade each item in the survey on a scale of 1–10 for clarity, appropriateness, relevance, length of the question, and the time required to complete it.

Any additional amendments recommended by the panel members were also discussed and considered. Finally, a pilot test was performed using the validated version of the questionnaire to assess the reliability and comprehensibility of the study questionnaire. The pilot test included 25 participants who were asked to complete the study questionnaire and report any questions or words that might hinder the understandability of the survey. The responses were imported into SPSS version 24 (IBM Corp, Armonk, NY, USA) software, and the internal consistency of the questionnaire items was calculated.

### 2.5. Statistical Analysis

After the questionnaire’s responses were collected, the raw data acquired were categorized and ordered using Microsoft Excel 2016. Data were then analyzed using SPSS 24 software. Via the use of descriptive statistics, categorical variables were described by means, frequencies, percentages, and 95% confidence intervals. Consequently, bivariate analysis involving using chi-square tests, Student’s *t*-tests, and one-way analysis of variance (ANOVA) were used to test for the significance of the association between categorical variables with a significance level of *p* < 0.05. When ordinal data were involved, Mann–Whitney U tests and Kruskal–Wallis tests were used. The level of statistical significance was set at *p* < 0.05. To investigate the strength of the association between the nominal variables, Phi and Cramer’s V (φc) where used: φc values between 0.05 and 0.10 were considered a weak relationship, whereas values between 0.10 and 0.15 were considered a moderately strong relationship, and values > 0.15 were indicative of a strong relationship [32].

## 3. Results

### 3.1. Demographic Characteristics

As shown in Table 1, most of the study subjects were female (*n* = 563, or 73.7%), and they were significantly greater in number as compared to the male respondents (one-sample chi-square test; χ^2^ (df = 1, *n* = 764) = 171.523, *p*-value < 0.001; 95% CI). Generally, the majority of participants were aged less than 25 years old (67.3%). The nationalities of the participants reported were 501 (65.6%) Arabic non-Emirati citizens, 173 (22.6%) Emirati, 59 (7.7%) Asian, 19 (2.5%) Western, 10 (1.3%) African, and 2 (0.3%) Latino. 

Of 764 participants, 606 (79.3%) were healthy, without any concomitant chronic medical conditions, while 158 (20.7%) had chronic comorbidities. The most prevalent chronic conditions, as shown in Table 2, were asthma and allergies, as well as metabolic diseases, such as diabetes, with a percentage of 5.6%, followed by cardiovascular diseases, including hypotension and hypertension 20 (2.6%). Around 0.5% of respondents suffered from either thyroid insufficiency, kidney disease, high cholesterol, or neurological/neuromuscular disease. Meanwhile, 0.3% of participants either had glucose-6-phosphate-dehydrogenase deficiency (G6PD) or psoriasis.

The efficacy of the Russian vaccine Sputnik V, which was measured in this questionnaire by rating the severity of symptoms if the patient became re-infected with COVID-19 despite receiving the vaccination as illustrated in Table 3, was not found to be related to the nationality/ethnicity of the vaccine recipient, as the *p*-value was not statistically significant (Pearson’s chi-square value; χ^2^ = 22.046, df = 20, *p*-value = 0.338; 95% CI). Variation was observed in the educational levels of the respondents. Almost three-quarters of the participants (76.5%) had a Bachelor’s degree as their level of education, while (9.4%) of the respondents stated that they had received high school education. Moreover, 6.5% held a higher diploma degree, and only (5.8%) reported that they attained a postgraduate education. The most prominent blood group types among the participants were group A+ with a percentage of 30.5% (*n* = 233), followed by O+ (*n* = 196, 25.7%), then B+ (*n* = 163, 21.3%). In Table 1, further demographic data of the participants can be visualized.

### 3.2. Assessment of Safety of COVID-19 Vaccines

When it comes to suffering from only one adverse effect of the vaccine, the most common singly reported side effects regardless the vaccination type were “pain at site of injection” with a percentage of 15.8% (*n* = 121), followed by “lethargy” (*n* = 30, 3.9%), then “general fatigue” (*n* = 29, 3.8%) as shown in Table 4. Interestingly, two of the side effects, i.e., nausea and vomiting, did not appear alone in any of the vaccine recipients. Instead, these side effects appeared in combination with other side effects. Figure 1 reveals that overall, nearly half of the vaccine recipients (*n* = 364, 47.6%) suffered from more than one adverse effect at the same time post vaccination. For instance, 49.1%, 28.5%, 23.4%, 17.4%, 13.4%, 13.1%, 11.5%, 5.2%, and 4.7% suffered from pain at injection site, lethargy, headache or migraine, muscle pain, dizziness, general fatigue, fever, diarrhea, and shortness of breath, respectively, along with other side effects, which are shown in Table 5.

The occurrence of side effects was not found related to the nationality/ethnicity of the vaccine recipient (chi-square test; χ^2^ = 79.051, df = 65, *p*-value > 0.01; 95% CI). The majority of the participants (*n* = 752 or 98.4%) denied that the vaccine caused any hypersensitivity reactions. Moreover, nearly three-quarters of the respondents (*n* = 579 or 75.8%) confirmed that they had been sleeping regularly after receiving the vaccine, and did not attribute sleeping irregularities to the vaccination. More data on the side effects of COVID-19 vaccines can be seen in Table 4. A chi-square goodness-of-fit test was conducted on the duration of the vaccines’ side effects; there were statistically significant differences regarding how long the side effects lasted (χ^2^ = 791.118, *p* < 0.001).

Figure 2 shows that a significant number of the respondents reported very slight (*n* = 276,40.4%) to mild (*n* = 164,25.3%) side effects for the Sinopharm vaccine (chi-square test; χ^2^ = 45.837, df = 20, *p* < 0.01; 95% CI). This is confirmed by the result of the mean score of the 6-point Likert scale question for Sinopharm vaccine recipients, which was found to be 2.02 with a standard deviation of 1.086. According to the result of Phi and Cramer’s V (φc = 0.172), which is > 0.15, there is a strong relationship between the number of post-vaccination side effects and the number of doses administered. A Kruskal–Wallis H test was conducted to investigate this relationship, as shown in Table 6, and it was found that most of the side effects occurred after the second dose of the vaccine; they were thus given a higher score on the 6-point Likert scale, indicating that that the vaccine side effect is more intense after the second dose (chi-square: χ^2^ = 238.109 df= 3, *p* < 0.001).

Using basic descriptive statistics, variation between males and females in terms of rating the severity of the vaccines’ side effects revealed that female participants had a higher mean score than males (2.11 vs. 1.74) Similarly using a Mann–Whitney test, it was found that females had a mean rank of 402.66, while males had a mean rank of 326.03, which is considered statistically significant as the *p*-value (two-tailed) was found to be <0.0001. Likewise, as presented in in Table 7, using a chi-square test, out of all the respondents, the females were found to be more susceptible to the adversities of COVID-19 vaccination (*p* < 0.001), with multiple (more than one side effect in one individual) side effects of slight severity (*p* < 0.001) (Figure 3).

## 4. Discussion

Shengli Xia et al. [33] proposed one of the first pieces of evidence that came out earlier in August 2020 confirming the efficacy of the BBIBP-CorV vaccine was obtained via the short-term analysis of two randomized clinical, which revealed that the Sinopharm vaccine is capable of stimulating immunogenicity with low incidences of adverse events. However, Phase III trials are required to further elaborate the vaccine’s efficacy in the long term. Based on the interim data concerning the Phase III clinical trials of the Sinopharm vaccine in Egypt, Jordan, Bahrain, and Peru, the vaccine’s efficacy was found to be 72.5% [34], which is lower than the 86% efficacy rate reported by the UAE in December 2020 [35]. Another country in which the vaccine has been trialed recently, in January 2021, is Brazil, and in a Brazilian study involving 12,000 health workers, it was found that the BBIBP-CorV vaccine is 78% effective at preventing mild cases of COVID-19 [36]. Moreover, a study conducted earlier in April 2021 by the University of Chile reported that the efficacy of two shots of the SinoVac after 2 weeks in Phase III trials had dropped to 56.5% [37]. However, the Sinopharm company have recently collaborated with the UAE to experiment an extra third dose, hoping to further increase the antibody response and enhance the efficacy [38]. Interestingly, a very recent observational study conducted in Bahrain involving a family that became reinfected despite receiving the vaccination concluded that the Sinopharm vaccine is able to offer protection and reduce casualty [39]. However, it cannot prevent the recurrence of the infection, and this finding seems to be due to a mutation in the S protein of the virus, referred to as E484K. Nevertheless, in the same study, two individuals who took different types of the vaccination, and had direct contact with an infected family member, did not show any symptoms. As a result, the Bahraini government started administering the Pfizer booster to Sinopharm vaccine recipients [39].

The findings of our study show that 94.4% of Sinopharm vaccine recipients have not been reinfected with COVID-19 after receiving two shots, taking into consideration that the majority of participants (*n* = 56,487.03%) received the vaccine doses in a period of time within January to February 2020. Moreover, 34 out of 36 participants who received the Sinopharm vaccine stated that they required no hospitalization upon contracting COVID-19 infection post vaccination, thereby suggesting that the vaccine can prevent hospitalization rates by 94.4%.

When it comes to the safety of the Sinopharm vaccine, B. Q. Saeed et al. [31] conducted a study earlier in April 2021 concerning self-reported side effects of SinoVac inside the UAE, and it was concluded that the first- and second-dose post-vaccination side effects were mild and predictable, and there were no hospitalization cases [31]. These data are in agreement with our study findings, which reveal that 92.9% of the participants reported very slight to moderately severe side effects experienced post vaccination, which tackles the vaccine-related conspiracy theories that claim the unsafety of the vaccine. On the other hand, the first piece of evidence obtained that confirmed the efficacy of the Pfizer-BioNTech COVID-19 vaccine was acquired from a randomized controlled trial (RCT) involving 43 thousand volunteers, who had a median age of 52 years old. The results revealed that the vaccine’s efficacy rate was around 95%, but it was also associated with several adverse events that took place within a few days of receiving the vaccination dose [40]_._ These adverse effects were divided into two categories, local or systemic side effects, and their severity ranged from mild to moderate [41]. According to the FDA report concerning the local side effects of Pfizer-BioNTech upon receiving the first dose and the second dose of the vaccine, it was found that the frequency of local side effects is slightly higher after the second dose in comparison with the first dose, and this trend was more significant in the case of systemic side effects [40]. Our study data confirm this trend only in the reported local side effect “pain at site of injection”, which was more frequent following the second dose of the vaccine, whereas the systemic side effect “lethargy” was more prominent following the first dose compared to the second dose.

According to Gushchin et al. [42] concerning the short-term efficacy of Sputnik V COVID-19 vaccine, those who had been vaccinated had no cases of moderate or severe COVID-19 infection after receiving the first dose of the vaccine for at least 21 days. Interestingly, our study findings reveal that only one out of the five recipients of the Sputnik V vaccine became reinfected with COVID-19 within 2–7 days of vaccination despite receiving the second dose. However, this participant revealed that the COVID-19 infection severity was very slight post vaccination. The majority of our study participants (*n* = 710 or 92.9%) stated that they did not become reinfected after receiving two doses of COVID-19 vaccine; therefore, the mean score of the Likert scale question regarding severity of the infection symptoms after vaccination was found to be 1.16 with a mean standard deviation of 0.660, which leans towards the answer “not been infected”.

Since the incidence of post-vaccination COVID-19 infections was significantly low, the incidence of being hospitalized after vaccination was negligible (*p* < 0.001). This was further confirmed by the fact that most individuals who became reinfected (*n* = 49 or 90.7%) stated that they did not require hospitalization during their reinfection period after the vaccine. However, it is noteworthy that about one-third of individuals reinfected with COVID-19 after vaccination (*n* = 18 or 33.33%) declared that they have caught the infection within a period of time that varied from 2–4 weeks from vaccination. In addition, when a Kruskal–Wallis H test was conducted, it showed a statistically significant difference in the severity of COVID-19 infection post vaccination between the different types of vaccination (chi-square: χ2(2) = 19.323, df= 4, *p* = 0.001), with a mean rank severity score of 376.63 for Sinopharm (Chinese), 427.38 for Pfizer (USA/Germany), 428.40 for Sputnik V (Russian), 355.50 for AstraZeneca/ University of Oxford, UK, and 415.73 for others. Out of the 764 participants involved in our study, consisting of 519 who do not consume any addictive substances and 245 smokers and those who consumed addictive substances such as caffeine and cigarettes, 93.1% of the non-smokers group (*n* = 483) experienced very slight to moderate symptoms of COVID-19 vaccines, whereas only 36 out of 519 reported severe symptoms. On the other hand, concerning the smokers group, 218 out of 245 had very slight to moderate symptoms post vaccination vs. 27 who reported severe symptoms. The odds ratio of developing severe vaccine symptoms in non-smokers vs. smokers was 1.662 (95% CI, 1.046 to 0.629). Additionally, a Mann–Whitney U test was ran to determine whether there were differences in the rating of the severity of COVID-19 symptoms in the case of recurrent infection post vaccination between smokers or consumers of addictive substances and non-smokers. Distributions of the engagement scores for both groups were similar, as assessed by visual inspection. The median engagement score was not statistically significantly different between the two groups, U = 61,807.5, z = −1.399, *p* = 0.162. Thus, our results reveal that the smoking status of cigarettes or consuming other addictive substances was found not to be related to the COVID-19 vaccination efficacy, as the reported (two-tailed) *p*-value was 0.162, which is not statistically significant. Using a chi-square test, out of all the respondents, our study reveals that females, especially younger individuals, belonging to the age group between 18–24 years old, were found to be more susceptible to the adversities of COVID-19 vaccination (chi-square: χ2(2) = 1824.174 df= 12, *p* < 0.001). This finding is in agreement with other studies in the literature that considered belonging to the female gender a significant risk factor for experiencing vaccination side effects, with a *p*-value of 0.0028 [43,44]. Al-Qazaz et al. attributed this increased incidence of multiple side effects post COVID-19 vaccination among females in particular to psychological and hormonal factors [45]. Another proposed explanation for this phenomena is the variability in the level of endogenous opioids and sex hormones between the two genders, which could lead to differences in the pain threshold and the extent of coping with stressors, according to Bartley et al. [46]. Lastly, numerous studies aiming to explain the probable reasons behind gender-related disparities in experiencing adverse events after COVID-19 vaccinations have reported that the female estradiol hormone tends to trigger the formation of more antibodies, which leads to more prominent immunological responses compared to men, whereas their testosterone sex hormone would act in an opposite manner, and result in the increased likelihood of contracting a viral infection due to the lowering the immune response [47,48,49].

## 5. Conclusions

In conclusion, the findings of our study are in line with other works in the literature in terms of the reported side effects of COVID-19 vaccines, which include local side effects, such as injection site pain, and systematic effects, including fatigue, fever, chills, and myalgia. In addition, most of these side effects occurred after the second dose of vaccine, irrespective of the type of vaccine. However, females, especially young individuals, were prone to experience more side effects. The majority of the respondents reported that these side effects began to appear within a day after vaccination, which is in agreement with other recorded literature studies. Nevertheless, since most of these side effects were tolerable, it can be concluded that COVID-19 vaccines available in the UAE are considered safe, and that they are recommended because they can prevent severe cases of COVID-19.

## 6. Strengths and Limitations of the Study

To our knowledge, this is the first study design dealing with the evaluation of the short-term efficacy and side effects of COVID-19 vaccines among the UAE population. The data provided may aid in decreasing the hesitancy to receive vaccination among the public. It is highly encouraged that further research is conducted concerning the vaccines’ safety in a way that clarifies the potential risk factors of experiencing intolerable side effects of the vaccines. Moreover, it is suggested that researchers perform prospective studies that include long-term follow-ups on vaccine recipients to attain a better understanding of the side effects. The main limitation of this study is the use of an online survey that was mainly based on participants’ self-reports. In addition to the unequal sample sizes taken from each emirate and the potential unintended gender bias in sampling, since more than half of the study subjects were female, as more of them responded to the questionnaire in comparison with male respondents. Another limitation of this study is that the number of participants over 44 years old is inadequate, which results in an age bias. Thus, one aspect of improvement that could be implemented in the future is to pay more attention to the older adult age groups when investigating efficacy and short-term safety of COVID-19 vaccines. Finally, we would like to highlight the importance of attempting to establish a sample size that is proportionally reflective of the real target population, since it would be a powerful addition to similar study designs (i.e., cross-sectional studies).

## Figures and Tables

**Figure 1 vaccines-10-02157-f001:**
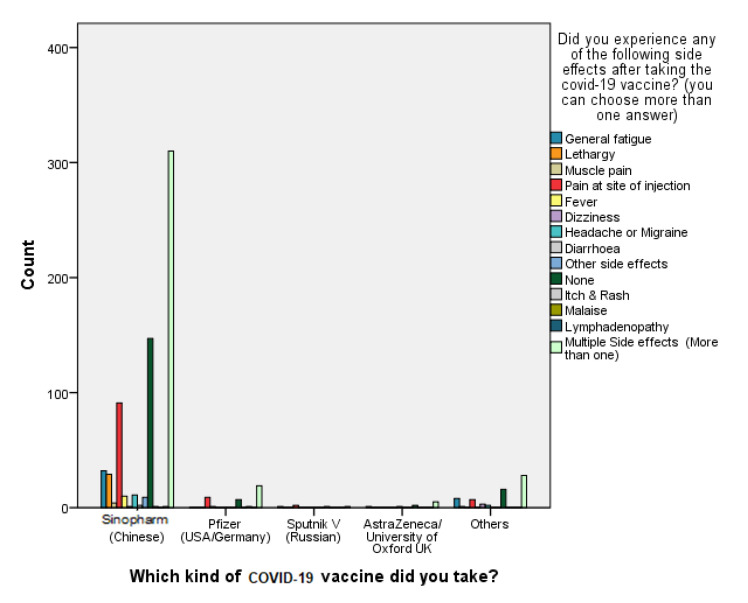
Reported side effects of different vaccines.

**Figure 2 vaccines-10-02157-f002:**
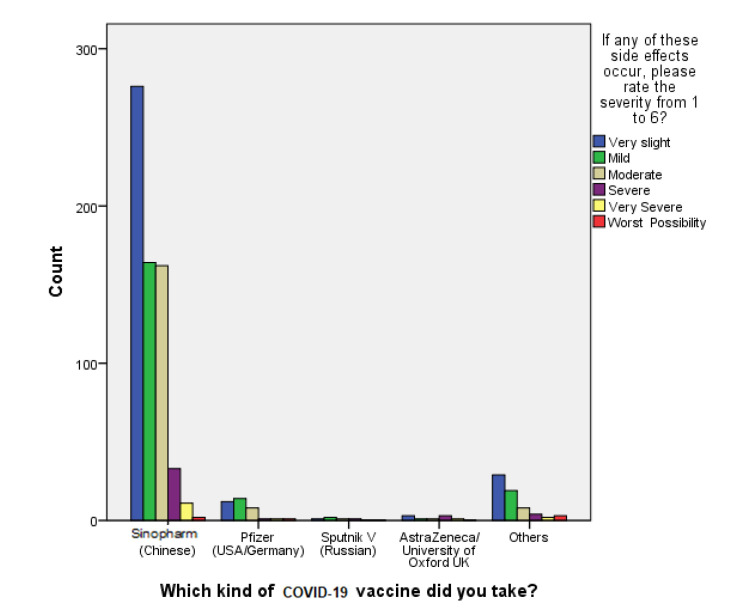
Reported severity of vaccines’ side effects.

**Figure 3 vaccines-10-02157-f003:**
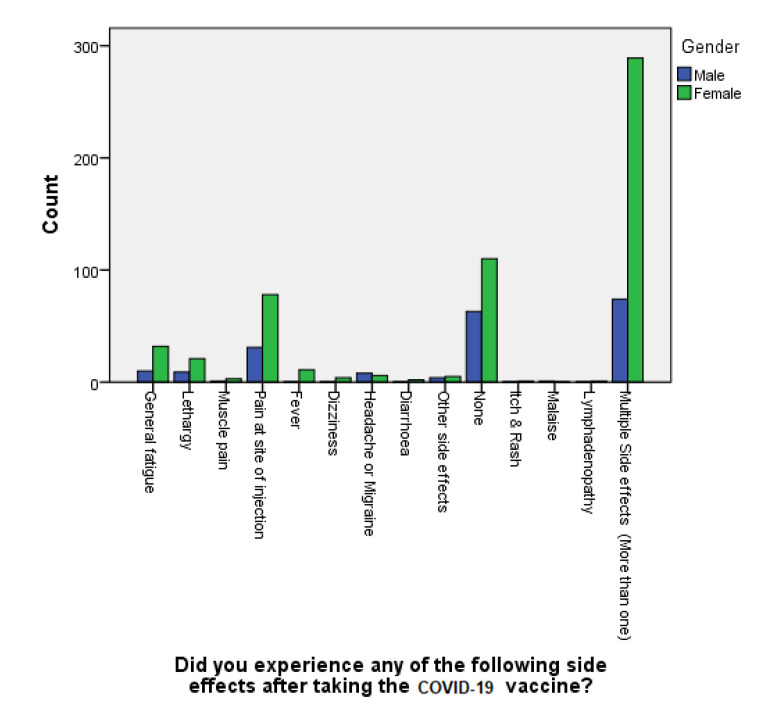
Male vs. female reaction to COVID-19 vaccine side effects.

**Table 1 vaccines-10-02157-t001:** Number and percentages of the questions on demographic information (*n* = 764).

Demographic	Groups	Frequency (*n*)	Percentages (%)
Age Group	18–24 years	514	67.3
	25–34 years	92	12.0
	35–44 years	87	11.4
	45–54 years	46	6.0
	55–64 years	14	1.8
	65 years and above	11	1.4
Gender	Male	201	26.3
	Female	563	73.7
Nationality	Arabic	501	65.6
	Emirati	173	22.6
	African	10	1.3
	Asian	59	7.7
	Western	19	2.5
	Latino	2	0.3
Educational level	Bachelor/University	585	76.6
	High School	72	9.4
	Higher Diploma	50	6.5
	Postgraduate (Masters/PhD)	44	5.8
	Middle School	12	1.6
	Non-Educated	1	0.1
Employment status	Unemployed	138	18.1
	Student	391	51.2
	Employed	224	29.3
City of residence	Sharjah	182	23.8
	Abu Dhabi	157	20.5
	Dubai	125	16.4
	Al-Fujairah	65	8.5
	Ajman	195	25.5
	Umm Al-Quwain	11	1.4
	Ras Al-Khaimah	29	3.8
Blood type	A+	233	30.5
	B+	163	21.3
	O+	196	25.7
	O−	46	6.0
	B−	20	2.6
	A−	24	3.1
	AB+	74	9.7
	AB−	8	1.0

Factors that might influence efficacy and short-term safety of COVID-19 vaccines.

**Table 2 vaccines-10-02157-t002:** Number and percentages of questions including factors that may be influencing the efficacy and short-term safety of the vaccines (total *n* = 764).

Question	Participants’ Answers	Frequency (*n*)	Percentages (%)
Do you suffer from one of the following chronic diseases or illnesses?	No chronic comorbidity	606	79.3
	Metabolic diseases (e.g., diabetes)	43	5.6
	Asthma or allergies	43	5.6
	Neurological or neuromuscular diseases	3	0.4
	Psoriasis	2	0.3
	Cardiovascular diseases (including hypertension and hypotension)	20	2.6
	Urticaria	1	0.1
	Anemia and thalassemia	1	0.1
	Cholesterol	3	0.4
	Sinus sensitivity	1	0.1
	Depression	1	0.1
	Thyroid insufficiency	3	0.4
	Glucose-6-phosphate-dehydrogenase deficiency	2	0.3
	Kidney diseases	3	0.4
	Seasonal sensitivity in nose, throat, or chest	1	0.1
	Migraine	1	0.1
	Liver disease	1	0.1
	Lung disease	1	0.1
	Other diseases	28	3.7
If you suffer from one of the previous chronic diseases, is your condition disease controlled?	I do not have any diseases	597	78.1
	The disease is not controlled	31	4.1
	The disease is controlled	136	17.8
Do you suffer from one of the following genetic or inherited diseases	None	726	95.0
	Sickle cell Anemia	4	0.5
	G6PD deficiency	8	1.0
	Thalassemia	12	1.6
	Hemophilia	1	0.1
	Other	13	1.7
How would you describe your body weight?	Overweight	265	34.7
	Normal	455	59.6
	Underweight	44	5.8
Which kind of COVID-19 vaccine did you take?	Sinopharm (Chinese)	648	84.8
	Pfizer (USA/Germany)	37	4.8
	Sputnik V (Russian)	5	0.7
	AstraZeneca/ University of Oxford, UK	9	1.2
	Other	65	8.5
Did you receive one dose (shot) or two doses of the COVID-19 vaccine?	One shot (one dose)	148	19.4
	Two shots (two doses)	616	80.6
In which month did you receive the first dose of the COVID-19 vaccine?	December 2020	64	8.4
	January 2021	427	55.9
	February 2021	212	27.7
	March 2021	17	2.2
	Other	44	5.8
Do you take addictive substances such as caffeine, cigarettes, or others?	No addictive substance consumption	519	67.9
	Smoking and consumption of addictive substances	245	32.1
Do you take any of the following medications? (You can choose more than one answer to this question)	I do not take any medications	604	79.1
	Aspirin and analgesics	56	7.3
	Contraceptives	7	0.9
	Antibiotics and antivirals	5	0.7
	Others	56	7.3
	Steroids	1	0.1
	Cancer treatment	4	0.5
	Antibiotics and antivirals + others	1	0.1
	Aspirin and analgesics + others	10	1.3
	Aspirin and analgesics + contraceptives	2	0.3
	Aspirin and analgesics + antibiotics and antivirals	5	0.7
	Aspirin and analgesics + steroids	1	0.1
	Contraceptives + others	1	0.1
	Aspirin and analgesics + contraceptives + others	1	0.1
	Aspirin and analgesics + antibiotics and antivirals + others	10	1.3

**Table 3 vaccines-10-02157-t003:** Number and percentages of the questions assessing efficacy of the vaccines (total *n* = 764).

Question	Participants’ Answers	Frequency (*n*)	Percentages (%)
Have you been infected with COVID-19 after receiving the second dose (shot) of the vaccine?	No	710	92.9
	Yes	54	7.1
If yes, did you need hospital admission?	Not been infected	710	92.9
	No hospitalization	49	6.4
	Hospitalized	5	0.7
When did you become infected with COVID-19 after receiving the vaccine?	Never infected	710	92.9
	After 2–4 weeks of vaccination	18	2.4
	After 2–7 days of vaccination	22	2.9
	After 1–3 months of vaccination	11	1.4
	After the day of vaccination	3	0.4
If yes, how severe was your case when were you infected?	1 = Not infected	710	92.9
	2 = Very slight	19	2.5
	3 = Mild	9	1.2
	4 = Moderate	17	2.2
	5 = Severe	9	1.2

**Table 4 vaccines-10-02157-t004:** Number and percentages of the questions assessing safety of the vaccines (total *n* = 764).

Question	Participants’ Answers	Frequency (*n*)	Percentages (%)
Did you experience any of the following side effects after receiving the COVID-19 vaccine? (Single choice only)	General fatigue	29	3.8
	Lethargy	30	3.9
	Muscle pain	4	0.5
	Pain at site of injection	121	15.8
	Chills	2	0.3
	Fever	3	0.4
	Dizziness	4	0.5
	Headache or migraine	14	1.8
	Diarrhea	2	0.3
	Other side effects	11	1.4
	Shortness of breath	1	0.1
	None	173	22.6
	Itch and rash	3	0.4
	Malaise	1	0.1
	Injection site swelling or redness	1	0.1
	Lymphadenopathy	1	0.1
	Multiple side effects	364	47.6
If any of these side effects occurred, please rate the severity from 1 to 6?	1= Very slight	321	42.0
	2= Mild	200	26.2
	3= Moderate	180	23.6
	4= Severe	42	5.5
	5= Very Severe	15	2.0
	6= Worst Possibility	6	0.8
Did you experience any side effects after the first dose or after the second dose?	None	281	36.8
	After 1st dose of vaccine	218	28.5
	After 2nd dose of vaccine	121	15.8
	After both doses of vaccine	144	18.8
When did the side effects start to appear after the vaccination?	Never occurred	248	32.5
	Within an hour of vaccination	195	25.5
	Within a day of vaccination	268	35.1
	Within a week of vaccination	47	6.2
	Within a month of vaccination	4	0.5
	Within more than a month of vaccination	2	0.3
How long did the side effects last?	0–24 h	440	57.6
	24–48 h	186	24.3
	2–7 days	91	11.9
	7–14 days	28	3.7
	More than 2 weeks	19	2.5
Did you experience any hypersensitivity after receiving the COVID-19 vaccine? Did you need hospitalization?	No	752	98.4
	Yes	12	1.6
Do you sleep regularly after receiving the vaccine?	Yes, regular sleep	579	75.8
	No, irregular sleep	185	24.2

**Table 5 vaccines-10-02157-t005:** Number and percentages of vaccination adverse events that are reported in combination with other side effects (multiple side effects occurring at the same time).

Question	Participants’ Answers	Frequency (*n*)	Percentages (%)
Did you experience any of the following side effects after receiving the COVID-19 vaccine? (You can choose more than one answer)	Pain at site of injection	375	49.1
	Lethargy	218	28.5
	Headache or migraine	179	23.4
	Muscle pain	133	17.4
	Dizziness	102	13.4
	General fatigue	100	13.1
	Fever	88	11.5
	Diarrhea	40	5.2
	Shortness of breath	36	4.7
	Malaise	33	4.3
	Injection site swelling or redness	30	3.9
	Chills	23	3.01
	Itch and rash	18	2.4
	Lymphadenopathy	6	0.8

**Table 6 vaccines-10-02157-t006:** Post-vaccination side effects severity rank according to number of doses.

Ranks			
	Did You Experience any Side Effects after the First Dose or after the Second Dose?	*n*	Mean Rank
If any of these side effects occurred, please rate the severity from 0 to 5?	None	281	235.12
	After 1st dose of vaccine	218	430.26
	After 2nd dose of vaccine	121	473.68
	After both doses of vaccine	144	521.16
	Total	764	
	Total	764	

**Table 7 vaccines-10-02157-t007:** Statistical results of chi-square tests (side effects, male vs. female).

Gender		Did You Experience any of the Following Side Effects after receiving the COVID-19 Vaccine?
Male	Chi-Square	276.716
	df	8
	*p*-value	0.0003
Female	Chi-Square	1824.174
	df	12
	*p*-value	0.0003

## Data Availability

Data will be available upon request.

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
