# Peer review of "Efficacy and Short-Term Safety of COVID-19 Vaccines: A Cross-Sectional Study on Vaccinated People in the UAE"

_vaccines, 2022, doi:10.3390/vaccines10122157_

Round 1

Reviewer 1 Report

I think that the study can contribute to the field.

1. Emphasizing the United Arab Emirates in the title of the study is successful.

2. The abstract of the study is inclusive.

3. I think that the methodology part is adequately explained.

4. It can be said that comprehensive information was given about the content of the questionnaire.

5. Questionnaire development and validation is explained in detail.

6. Information on statistical analysis is also given.

5. Emphasizing the strengths and weaknesses of the study is successful.

I had the impression that the study was explained in detail. For these reasons, I find it appropriate to publish the study in this form. Best wishes, best regards.

Author Response

We would like to thank the author foe his valuable comments and the impression that the study was explained in detail.

We appreciate your time reading our study

Thanks again

Reviewer 2 Report

Dear authors, the presented work is interesting. It comprehensively describes the short-term adverse effects and efficacy of COVID-19 vaccines. All article sections are detailed and methodologically presented. I would encourage including potential bias regarding the age of participants and gender. 

1. What is the main question addressed by the research?

The main question is short term adverse reactions to vaccines and the short-term effectiveness of vaccination
2. Do you consider the topic original or relevant in the field? Does it
address a specific gap in the field? The topic is relatively original and relevant.
3. What does it add to the subject area compared with other published
material? The research provides additional information on the short-term effects of vaccination primarily administered among youths.
4. What specific improvements should the authors consider regarding the
methodology? What further controls should be considered? No improvements are required. Further controls are not necessary. 
5. Are the conclusions consistent with the evidence and arguments presented
and do they address the main question posed? The conclusions are consistent with provided information in the article.
6. Are the references appropriate? The references are appropriate
7. Please include any additional comments on the tables and figures. There are no additional comments. The article is well-structured and written.

Author Response

We would like to thank the author for his valuable comments and suggestions.

Please note that the comment "I would encourage including potential bias regarding the age of participants and gender “ is addressed in the STRENGTH AND LIMITATION section in the attached manuscript (Page 20, line 490-497). Please see the attached manuscript.

Thank you again for your time reviewing our work

Best wishes

Reviewer 3 Report

The manuscript described efficacy and short-term safety of COVID-19 vaccines. The authors showed vaccines can offer a short-term protection of COVID-19 reinfection. Thus, these findings will be useful for the treatment of COVID-19. Therefore, the manuscript is not too excellent to be published after revision. In other words, the manuscript is so excellent that it should be published after revision.

Comments

(1) Did the efficacy of a Russian vaccine depend on race or not?

(2) In this study, were there the different results between male and female? Multiple side effects were observed especially in female in Figure 3? Why?

(3) Was smoking involved in the efficacy of COVID-19 vaccines?

(4) In Table 2., “I don't have any diseases” should be replaced with “I do not have any diseases”.

That is all.

Author Response

First of all, we would like to express our deep gratitude to the reviewer for sharing his valuable suggestions and insights on the manuscript.

In response to your much-respected comments, we addressed these comments as per the following:

Comment (1): Did the efficacy of a Russian vaccine depend on race or not?

Response 1: Please note that this comment is addressed on (page 8, line 248) under the section (Results-Demographic characteristics); a paragraph starting with “The efficacy of the Russian vaccine Sputnik V which is measured in this questionnaire via…” was added to address this comment, and we conducted Pearson Chi-square statistical test which revealed non-significance of the race on the vaccine’s efficacy.

Comment (2): In this study, were there the different results between male and female? Multiple side effects were observed especially in female in Figure 3? Why?

Response 2: Please note that this comment is addressed on (page 19, line 443) under the section (Discussion); a paragraph starting with “Al -Qazaz et al., have attributed this increased incidence …” was added to address this comment. The question was answered with the help of reviewing some of the recently published research that aims to address the gender-based dissimilarities in the side effect profile post COVID-19 vaccination

Comment (3): Was smoking involved in the efficacy of COVID-19 vaccines?

Response 3: Please note that this comment is addressed on (page 19, line 432) under the section (Discussion); a paragraph starting with “Additionally, A Mann-Whitney U test was run to determine if…” was added to address this comment, and we conducted Mann-Whitney U test statistical test which revealed non-significance of the smoking status on the vaccine’s efficacy.

Comment (4): In Table 2., “I don't have any diseases” should be replaced with “I do not have any diseases”.

Response 4: Please note that this comment is addressed on (page 10) in Table 2.
